# Effect of Melanization on Thallus Microstructure in the Lichen *Lobaria pulmonaria*

**DOI:** 10.3390/jof8080791

**Published:** 2022-07-28

**Authors:** Amina G. Daminova, Alexey M. Rogov, Anna E. Rassabina, Richard P. Beckett, Farida V. Minibayeva

**Affiliations:** 1Kazan Institute of Biochemistry and Biophysics, FRC Kazan Scientific Center, Russian Academy of Sciences, P.O. Box 261, 420111 Kazan, Russia; daminova.ag@gmail.com (A.G.D.); aerassabina@yandex.ru (A.E.R.); 2Interdisciplinary Center for Analytical Microscopy, Kazan (Volga Region) Federal University, 420018 Kazan, Russia; alexeyrogov111@gmail.com; 3School of Life Sciences, University of KwaZulu-Natal, PBag X01, Scottsville 3209, South Africa; rpbeckett@gmail.com

**Keywords:** lichens, melanins, mycobiont, scanning electron microscopy, transmission electron microscopy, UV stress

## Abstract

Lichens often grow in microhabitats where they experience severe abiotic stresses. Some species respond to high UV radiation by synthesizing dark brown melanic pigments in the upper cortex. However, unlike the melanized structures of non-lichenized fungi, the morphology of the melanic layer in lichens remains unstudied. Here, we analyzed the morphology, ultrastructure, and elemental composition of the melanized layer in UV-exposed thalli of the lichen *Lobaria pulmonaria* (L.) Hoffm. Using light microscopy, we detected a pigmented layer sensitive to staining with 3,4-L-dihydroxyphenylalanine, a precursor of eumelanin, in the upper cortex of melanized thalli. Analysis of cross-sections of melanized thalli using scanning electron microscopy revealed that melanin-like granules are deposited into the hyphal lumens. Melanized thalli also possessed thicker hyphal cell walls compared to pale thalli. Energy-dispersive X-ray spectroscopy analysis of the elemental composition of the hyphal walls and extracted melanin indicated that the type of melanin synthesized by *L. pulmonaria* is eumelanin. Transmission electron microscopy was used to show that during melanization melanosome-like dark vesicles are transported to the cell surface and secreted into the cell walls of the fungal hyphae. Results from this study provide new insights into the effects of melanin synthesis on the microstructure of lichen thalli.

## 1. Introduction

Lichens are ancient symbiotic photosynthesizing organisms, and it has been suggested that they were one of the first to colonize the terrestrial biosphere. Today, lichens still dominate in large areas of terrestrial ecosystems with harsh climates. They can grow on almost any substrate, including soil, rocks, and tree bark [1]. The lichen thallus is a symbiotic system formed by a mycobiont (fungal partner, mainly ascomycetes) and a photobiont (algae or cyanobacteria) [2]. Lichens can successfully withstand severe abiotic stresses such as dehydration, temperature extremes when desiccated, and exposure to ultraviolet (UV) radiation [3]. Some lichens respond to elevated UV by synthesizing secondary metabolites, including the dark pigment melanin, in the upper cortex [4]. The term melanin comes from the Greek word “melanos” meaning black or dark-colored. The widespread presence of melanins in a great diversity of organisms suggests that melanogenesis is evolutionarily ancient, and therefore important for life [5]. Melanins are hydrophobic heterogeneous biopolymers formed by sequential oxidative polymerization of phenolic or indole compounds. There are different types of this pigment including eumelanin, pheomelanin, neuromelanin, allomelanin, and pyomelanin. The melanins present in lichens are mainly eumelanins and allomelanins [6]. Although the different types of melanins differ in their chemical structure, nevertheless, in most organisms, they share common functions such as thermoregulation, radical scavenging, energy conversion, and immunity formation [7]. Melanins can absorb radiation from a wide range of wavelengths, including γ-rays, X-rays and UV, allowing them to have a photoprotective function [8]. Despite the increasing awareness that the properties of melanins are dependent not only on their chemical composition, but also their ultrastructure and localization [9], surprisingly little is known about the architecture of melanic thallus structures in lichens.

The main aim of the work presented here was to study the morphology, localization, and elemental composition of the melanized upper cortex of the foliose lichen *Lobaria pulmonaria* (L.) Hoffm exposed to UV. *Lobaria pulmonaria* is a large epiphytic lichen consisting of an ascomycete fungus and a green algal partner (*Symbiochloris*) living together in a symbiotic relationship with a cyanobacterium (*Nostoc*). The cyanobacteria occur in small (0.5–1.5 mm in diameter) pockets on the lower surface of the thallus, and are termed cephalodia [10]. Recent studies have suggested that this species is particularly vulnerable to climate change and modern forestry practices [11]. The species has a wide distribution in Africa, Asia, Europe, and North America, preferring shaded, damp habitats with high rainfall, especially in coastal areas [12]. The average light level throughout the entire year for the habitat where *L. pulmonaria* typically grows can be as low as 14 μmol m^−2^ s^−1^, although it is ten times higher on the exposed side of tree trunks [13]. However, at times, the lichen may be exposed to higher light levels, for example, following leaf fall in autumn. *Lobaria pulmonaria* also receives more light when openings in forests occur, for example, because of tree fall, or where local conditions such as avalanches, poor soils, or fire damage create semi-permanent clearings. The most important adaptation to increased exposure to high solar radiation, and particularly UV is probably the synthesis of melanin pigments in its upper cortex [8]. It is therefore readily possible to compare pale with melanized thalli, making *L. pulmonaria* a good model species to investigate the architecture of the melanin-containing upper cortex using a diverse array of microscopy techniques.

## 2. Materials and Methods

### 2.1. Materials

The lichen *Lobaria pulmonaria* was collected from the bark of oak trees at Langangen, South Norway (59.096 N, 9792 E). Pale, lightly melanized and more strongly melanized thalli were collected growing close to each other, with the lightly and more heavily melanized thalli occurring in progressively more exposed microhabitats. Material was slowly dried at room temperature at 60–70% relative humidity and stored at −20 °C until use.

### 2.2. Methods

#### 2.2.1. Reflectance Measurements

Reflectance spectra of the upper cortex were recorded on dry, intact lichen discs. Browning reflectance indexes of the lichen thalli were assessed using the integrating sphere method, as described in Beckett et al. [14]. Briefly, an integrating sphere (model ISP-50-REFL, OceanOptics, Eerbeek, The Netherlands) was pressed against the thalli, which were then illuminated with a halogen lamp (model DH2000, OceanOptics) through a 600 µm thick optical fibre connected to the input port of the integrating sphere. Reflectance (400–1050 nm) was measured with a spectrometer (model SD2000, OceanOptics) connected to the output port of the sphere with a 400 µm fibre. Reflection was calculated relative to a reference spectrum derived from a white reference tile (WS-2, OceanOptics). By using the readings at 550, 700 and 750 nm, the browning reflectance index (BRI), was calculated as BRI = (1/R550 − 1/R700)/R750 [15], and this used as a quantitative estimate of melanic compounds. To analyze the data statistically, one way ANOVA was carried out, followed by Duncan’s multiple range test. Columns with different letters on top differ significantly (*p* < 0.05).

#### 2.2.2. Light Microscopy

Before the start of the experiments, lichens were hydrated in a chamber at 10 °C on moist filter paper at 100% relative humidity and dim light for 24 h. To visualize melanin, thalli were embedded in 3% agarose blocks and 50 µm sections were cut using a vibratome (Leica VT 1000S, Wetzlar, Germany). For qualitative reactions, three types of staining were applied. First, sections were stained with 0.2% 3,4-L-dihydroxyphenylalanine (L-DOPA) for 1 h in the dark [16]. Second, the sections were stained with 1% azure ΙΙ solution for 30 min and then 0.5% eosin solution for 1–2 min, according to the Rejniak protocol [17]. Third, the sections were incubated in 2.5% FeSO_4_ for 1 h followed by incubation in 1% K_3_[Fe(CN)_6_] for 30 min, and then in an ethanol-xylene mixture (50:50 by volume), according to the Lillie protocol [18]. The sections were analyzed using an epifluorescence microscope, the Leica DM1000 (Leica Biosystems, Wetzlar, Germany). Images were obtained using a digital camera at a magnification of ×40. For determination of the thickness of the pigmented layer, 15 cross sections from three biological replicates were analyzed using the built-in microscope software.

#### 2.2.3. Scanning Electron Microscopy

The sections of non-melanized and melanized lichen thalli were fixed in 2.5% glutaraldehyde in 0.1 M Na-phosphate buffer pH 7.4 overnight, then dehydrated in 30, 40, 50, 60, 70, 80, 90, and 96% ethanol. The sections were then incubated in a mixture of hexamethyldisiloxane (HMDS) and ethanol in ratios of 1:3, 1:1, 3:1 (by volume) for 30 min and 100 % HMDS for 60 min. The specimens were mounted on aluminum stubs with double-sided carbon tape and sputter-coated with gold using the Q150T ES coater (Quorum Technologies, Lewes, UK). The morphology of the structured surface of the non-melanized and melanized lichen thalli was observed using a high-resolution scanning electron microscope (SEM), Merlin (Carl Zeiss, Oberkochen, Germany) at an accelerating voltage of 5 kV, and the elemental composition of melanized samples was determined using an Energy-dispersive X-ray (EDX) detector, AZtec X-max (Oxford Instruments, Oxford, UK) fitted to the microscope at an accelerating voltage of 20 kV. The line intensities for each element of the sample and for the same elements in the calibration standards with known composition were measured [19]. Up to 500,000 impulses were accumulated. The results were normalized up to 100%. To assess the relative thickness of hyphal cell walls, the number of hyphae in thalli per unit of the cross-sectional area on ten SEM images was compared between pale and melanized thalli.

#### 2.2.4. Transmission Electron Microscopy

Sections of non-melanized and melanized lichen thalli were fixed in 1% glutaraldehyde in 0.1 M Na-phosphate buffer pH 7.4 overnight at 4 °C and in 2.5% glutaraldehyde for 4 h at room temperature. After washing in Na-phosphate buffer, the samples were post-fixed by incubation in 1% osmium tetroxide prepared in 0.1 M Na-phosphate buffer pH 7.4 for 1 h. Then, samples were dehydrated in a graded aqueous ethanol series, transferred to acetone, and immersed in LR White resin (Medium Grade Acrylic Resin; Ted Pella, Redding, CA, USA) that contained acetone added in the proportions (*v*/*v*) 1:3, 2:3, 3:1, with each step involving a 24 h incubation. The samples were then embedded in LR White resin in Beem capsules and polymerized at 60 °C for 24 h. Ultra-thin sections (c. 100 nm thick) were cut using an ultramicrotome (LKB-8800, Bromma, Sweden) and mounted on copper grids. The sections were stained with 2% aqueous uranyl acetate (*w*/*v*) for 20 min and Reynolds’ lead citrate for 7 min [20]. Finally, the sections were examined using a Hitachi HT 7700 Excellence transmission electron microscope (TEM, Tokyo, Japan) at an accelerating voltage of 100 kV.

#### 2.2.5. Extraction and Purification of Melanin

Lichen samples were ground to a powder with the addition of liquid nitrogen. The resulting powder was mixed with 2 M NaOH, pH 10.5. In preliminary experiments it was found that melanin was not extracted by water, ethyl acetate, acetone, and chloroform [6]. After 24 h incubation, the mixture was filtered and then centrifuged (Hermle Z36-HK, Gosheim, Germany) at 15,000× *g* for 10 min. The supernatant was acidified by adding 2 M HCl to pH 2.5, incubated at room temperature for 12 h, and then centrifuged at 15,000× *g* for 10 min. The resulting precipitate was rinsed with distilled water and, subsequently, rinsed with organic solvents such as chloroform, ethyl acetate and acetone, and after, oven dried at 50 °C. The purified melanin was presented as dark brown powder without foreign inclusions.

## 3. Results

Cross-sections of non-melanized (Figure 1A,B) and melanized (Figure 1F,G) *Lobaria* thalli were obtained to visualize the upper cortex. Cephalodia were present in all thalli examined (images not shown). In *L. pulmonaria*, the upper cortex is generally considered to be comprised of paraplectenchyma [1]. In melanized thalli, a specific layer of pigmented cells occurred in the upper cortex; beneath this layer was the algal layer, containing some unpigmented fungal hyphae (Figure 1G). The thickness of the pigmented layer was on average 20 ± 2.8 µm. To confirm the melanized nature of the pigmented cells, sections of *Lobaria* thalli were stained with L-DOPA, a precursor of eumelanin. Following staining, the upper layer of the cortex became intensively brown in melanized, but not pale, thalli (Figure 1C,H). Using the Rejniak assay, black granules were visualized in the cells of the melanized upper cortex (Figure 1I). After staining, using the iron oxidation-based Lillie protocol, a dark pigment was also visualized only in the cortex of melanized thalli (Figure 1J).

Samples of pale, weakly melanized and strongly melanized lichen thalli were selected based on their browning indexes (Figure 2) and examined using SEM. The non-melanized upper cortical layer comprised fungal hyphae with empty lumens and a smooth inner surface (Figure 3A). In cross-sections of weakly melanized thalli, the internal surface of the lumens of fungal hyphae had a rough appearance (Figure 3B, white arrow). The upper cortical layer of strongly melanized thallus was characterized by the deposition of metabolites, including spherical melanin-like granules, in the hyphal lumens (Figure 3C,D black arrows). Compared to pale samples, the hyphal cell walls in melanized thalli were visibly thicker (Figure 3). Moreover, the number of hyphae in pale thalli per unit of the cross-sectional area was 1.5 and 2.5 times greater than in weakly and strongly melanized thalli, respectively.

The elemental composition in pale and strongly melanized thalli of *L. pulmonaria* was analyzed by EDX in three selected zones of the paraplectenchyma that form the upper cortical layer: zone 1 with inclusions or melanin-like granules within the lumen of the hyphae; zone 2 from the rough internal surface of the lumen of the hyphae; zone 3 from hyphal cell walls (Figure 4).

In both thalli, all zones were mostly comprised of C, N, and O (Table 1). Interestingly, the relative weight of N in zone 1 with melanin-like granules (Figure 4B) was 1.6 times greater in a melanized thallus compared to that in zone 1 in a pale thallus (Figure 4A). The C to N ratio in the melanin granules was c. 11 (Table 1). Si was also present in all selected zones, but in melanin granules the relative weight was on average 1.3 times greater than in the inner surface of the lumen of a pale thallus. Other elements were detected in minor quantities.

Analysis of the elemental composition of extracted and purified melanin from *L. pulmonaria* by EDX confirmed the ratio of C/N as 11. Other elements were also present in minor quantities (Figure 4C).

Images of non-melanized thalli (Figure 5A,B) and stages of melanization were visualized in ultra-thin cross-sections of thalli of *L. pulmonaria* using TEM. During the initial stages, melanosome-like dark vesicles accumulated in the fungal hyphae (Figure 5C). Later these vesicles were transported to the cell surface where they merged with the plasma membrane (Figure 5D), aggregated, and became imbedded in the cell wall (Figure 5E,F). The electron density of the cell wall significantly increased as a result of melanin secretion (Figure 5E,F).

## 4. Discussion

The lichen *L. pulmonaria* responds to solar radiation by synthesizing the brown pigment melanin in the paraplectenchyma that forms the upper cortex of the thallus [4] (Figure 1). Browning can be quantified by measuring the BRI index [14], which in the present study was on average c. 15 in the more heavily melanized thalli (Figure 2). This value is similar to our earlier measurements of *L. pulmonaria* from the same site [21], although sometimes even darker thalli can be collected with BRIs exceeding 40 [22]. Perhaps surprisingly, there have been few attempts to study the structure of melanized thallus parts in lichens. Elucidation of their microstructure and ultrastructure will enable us to understand how the pigment is assembled and how it interacts with other cellular components. Furthermore, knowing the precise cellular location of the pigment will provide insights into its biological role. Results presented here suggest that in the upper cortex of *L. pulmonaria* melanization is accompanied by the deposition of spherical melanin-like granules in the lumens of the hyphae and significant changes in the microstructure of the cell wall (Figure 3). The high N content of the melanin-like granules in hyphae measured using EDX suggested that in *L. pulmonaria* the type of melanin synthesized is eumelanin. It seems likely that morphological changes in the upper cortex of melanized thalli, such as the thickness and roughness of the hyphal cell walls, may result in a stiffening and hardening of the thallus, which may increase stress tolerance.

In the lichen thallus, the upper cortex of the mycobiont forms a protective “matrix” that surrounds the photobiont. The thickened cell walls of the mycobiont form a network that covers the algal or cyanobacterial cells [2]. In some lichen species, exposure to high solar radiation, particularly UV, induces the formation of melanic pigments [4]. In fungi, melanins are considered to play a variety of roles in stress tolerance [23]. Specifically in lichens, recent findings suggest that their main roles are to protect the mycobiont against UV and the photobiont from photoinhibition [4]; they also possess strong antioxidant properties [6]. It is becoming clear that the properties and functions of melanins are determined by their ultrastructural characteristics and chemical composition [9]. The ultrastructure of melanins has been best studied in pathogenic fungi [24], pathogenic bacteria [25], and human melanosomes [26,27]. Detailed analyses by TEM and SEM have demonstrated that the melanins from pathogenic fungi and humans have a granular structure [24,28]. For example, in the fungal pathogen *Cryptococcus neoformans* melanin nanoparticles aggregate into large melanin granules, which then strongly bind to components of the fungal cell wall [29]. In the present study, hyphae in the upper cortex of strongly melanized thalli of the lichen *L. pulmonaria* were found to possess spherical melanin-like granules in their lumens (Figure 3C,D black arrows). Granules could be up to 500 nm, similar in size to those reported from a pathogenic fungus [28]. Such granules were absent in non-melanized and weakly melanized thalli (Figure 3 A,B). Therefore, the morphology of melanins in lichenized fungi resembles that of other organisms.

The C to N ratio in the melanin granules (Figure 4B) was c. 11 (Table 1), suggesting that the type of melanin synthesized by this lichen is eumelanin [30]. These data are consistent with the elemental composition of extracted and purified melanin from *L. pulmonaria* (Figure 4C). Earlier, we found the C/N ratio of melanin from *L. pulmonaria* to be 10.4 by using a CHNS-O Elemental Analyzer (Eurovector SpA, Redavalle, Italy), following burning of the extracted and purified melanin in the presence of an oxidizer in an inert gas [31,32]. It seems likely that the presence of N-fixing cephalodia allows *L. pulmonaria* to synthesize N-rich eumelanins, rather than the N-poor allomelanins reported from non-N-fixing lichens [6], although whether eumelanins offer superior stress protection, e.g., from UV light, remains unclear [4].

While melanins can occur as intracellular granules, in fungi they can also be secreted to the external environment and accumulate in cell wall pores, where they form cross-links with cell wall polysaccharides [33]. For example, a comparative analysis of *Fonsecaea pedrosoi* cells cultivated with or without the DHN melanin-specific inhibitor tricyclazole indicated that melanin is involved in cross-linking cell wall components [28]. It is probable that diverse additional constituents are involved in the localization and maintenance of melanin within the complex cell wall structure. For example, melanins can form complexes with components of the mycobiont cell wall such as chitin, a polymer consisting of N-acetylglucosamine and glucosamine units linked by β(1,4) covalent bonds [34]. Melanin production in the fungal pathogen *Candida albicans* was boosted by the addition of N-acetylglucosamine to the medium, indicating a possible association between melanin production and chitin synthesis. It is possible that as in *C. albicans*, in *L. pulmonaria* melanin also binds to chitin in the cell walls. Interestingly, melanization is associated with changes in the microstructure of the hyphae. In melanized thalli, the surfaces of the lumina are rougher (Figure 3B,C), and the cell walls are twice as thick as those of non-melanized thalli (compare Figure 3A with Figure 3C,D).

In fungi, early studies suggested that melanin is synthesized inside cellular vesicles analogous to mammalian melanosomes [35]. Later, evidence was provided that the precursors of melanins are also synthesized inside such vesicles [36], and melanins are then secreted into the cell wall. In the present study, the stages of melanization visualized in *L. pulmonaria* are similar to those suggested by a study carried out by Camacho et al. (2019) on the pathogenic fungus *C. neoformans* [29]. During melanization of the lichen *L. pulmonaria*, melanosome-like dark vesicles accumulate in the cells of the fungal hyphae (Figure 5C), are then transported to the cell surface where they merge with the plasma membrane (Figure 5D), and finally are secreted into the cell wall (Figure 5E,F). As a result of the binding of melanin nanoparticles to the components of the cell wall and their subsequent aggregation, spherical melanin-like granules appear in the lumens of the fungal hyphae. Therefore, results presented here suggest that in lichenized fungi, melanogenesis involves a sequence of intracellular events similar to those that occur in pathogenic fungi.

## 5. Conclusions

Taken together, results from this study provide new insights into the changes in the microstructure of lichen thalli that are associated with melanin synthesis. We discovered that, as for some non-lichenized fungi, in lichens melanization is accompanied by the formation of melanin-like granules in the upper cortex. Other structural changes in the morphology of the cortical layer associated with melanization include an increased thickness of the cell walls of the hyphae that may result in a stiffening and hardening of the thallus. All these changes in the microstructure of thalli may reduce the transmittance of UV or high levels of photosynthetically active radiation, increasing the tolerance of both the myco- and photobiont to stress.

## Figures and Tables

**Figure 1 jof-08-00791-f001:**
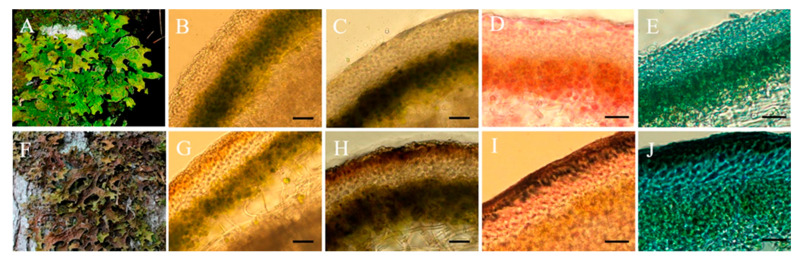
Non-melanized (**A**–**E**) and melanized (**F**–**J**) Lobaria pulmonaria thalli. Whole thalli (**A**,**F**). Cross sections of thalli: unstained (**B**,**G**), DOPA reaction (**C**,**H**), Rejniak staining (**D**,**I**), Lillie staining (**E**,**J**). Darkly stained melanins are clearly visible in the upper cortex of melanized thalli. Bar = 25 µm.

**Figure 2 jof-08-00791-f002:**
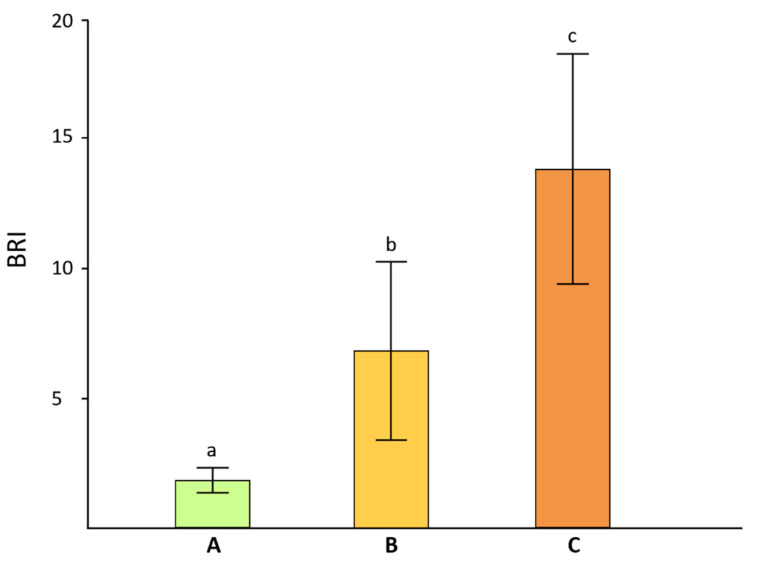
Browning reflectance index of non-melanized (**A**), weakly melanized (**B**) and strongly melanized (**C**) lichen thalli. Columns with different letters on top differ significantly (*p* < 0.05).

**Figure 3 jof-08-00791-f003:**
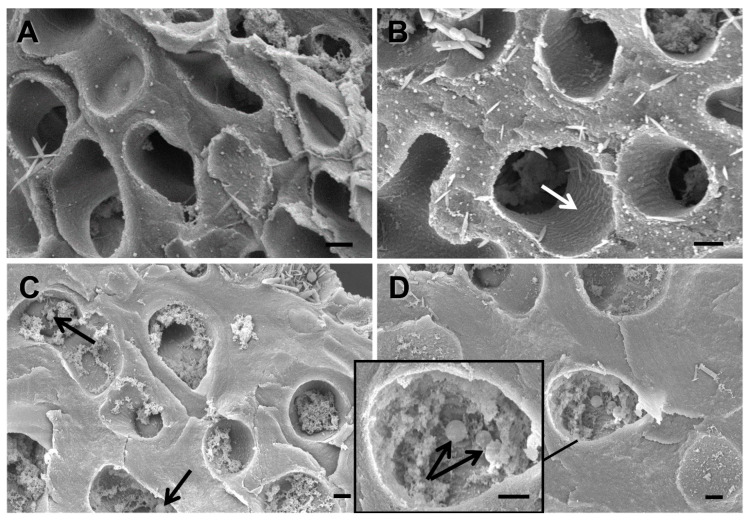
SEM images of cross sections from non-melanized (**A**), weakly melanized (**B**) and strongly melanized (**C**,**D**) *Lobaria* thalli. White arrow indicates the rough appearance of internal surface of lumens of fungal hyphae. Black arrows indicate spherical melanin-like granules. Bar = 1 µm.

**Figure 4 jof-08-00791-f004:**
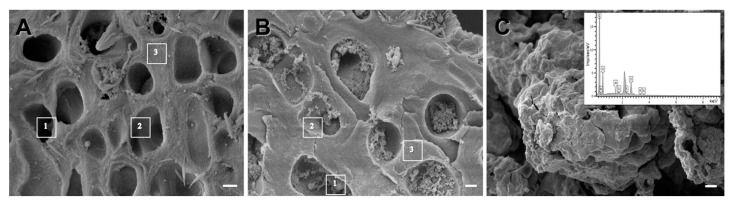
SEM image of cross sections from non-melanized (**A**) and melanized (**B**) *Lobaria* thalli with three areas selected for elemental composition analysis: (**A**)—Lumen of hyphae (1), Lumen of hyphae with smooth surface (2), Hyphal cell walls (3); (**B**)—Lumen of hyphae with melanin granules (1), Lumen of hyphae with rough surface (2), Hyphal cell walls (3). SEM image of extracted and purified melanin from *Lobaria pulmonaria* (**C**) and EDX spectrum of elemental composition (insert). Bar = 1 µm.

**Figure 5 jof-08-00791-f005:**
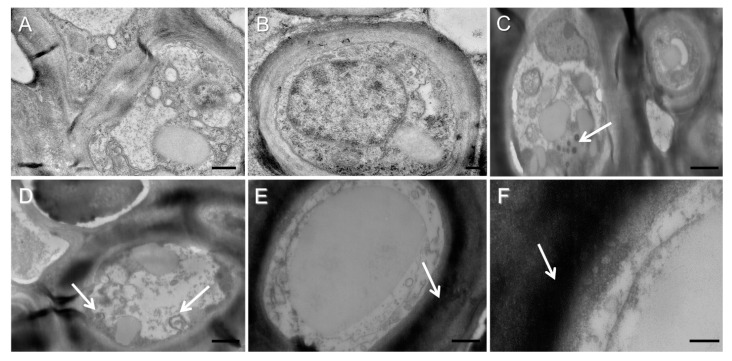
TEM images of non-melanized thalli (**A**,**B**) and the stages of melanization in melanized thalli of *Lobaria pulmonaria* (**C**–**F**): (**C**)—accumulation of melanosome-like vesicles. (**D**)—merging vesicles with plasma membrane. (**E**,**F**)—electronically dense cell walls in melanized thalli. Arrows indicate melanin-like vesicles. Bar = 0.5 µm (**A**), 0.2 µm (**B**), 1 µm (**C**,**D**), 0.5 µm (**E**), 0.2 µm (**F**).

**Table 1 jof-08-00791-t001:** Elemental composition (in %) of three zones in non-melanized and melanized cortical layer of *L. pulmonaria*.

Zones	C	N	O	Si	Al	Na	S	Cu
**Non-melanized thallus of *L. pulmonaria***								
Lumen of hyphae (1)	68.6	3.4	21.4	3.2	0.1	1.9	0.6	0.6
Lumen of hyphae with smooth surface (2)	65.7	4.9	24.8	3.0	0.2	1.4	0.6	1.0
Hyphal cell walls (3)	63.5	5.0	26.3	3.2	0.2	1.3	0.7	1.0
**Melanized thallus of *L. pulmonaria***								
Lumen of hyphae with melanin granules (1)	60.8	5.5	27.6	4.1	0.1	0.9	0.3	0.7
Lumen of hyphae with rough surface (2)	64.2	4.3	26.3	3.8	0	0.8	0.3	0.4
Hyphal cell walls (3)	64.7	3.9	28.0	3.6	0.1	1.0	0.3	0.5

## Data Availability

Not applicable.

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
