# Peer review of "Effect of Melanization on Thallus Microstructure in the Lichen Lobaria pulmonaria"

_jof, 2022, doi:10.3390/jof8080791_

Round 1

Reviewer 1 Report

Effect of melanization on thallus microstructure in the lichen  Lobaria pulmonaria

Amina Daminova1, Alexey Rogov2, Richard Peter Beckett3 and Farida Minibayeva1,2,*

The topic is very interesting. The paper is very well written, presenting a less known field within lichenology in a complex way via the application of various methods. The introduction contains clear objectives and detailed explanations to the hypothesis. Methods contain the most important information. Results are presented by the shortest possible text, however, together with the illustrations added, the presentation of the results is excellent, easy to follow. Chapters Discussion and Conclusion are clear as well. Congratulations to the study and the results.

Some minor mistakes are indicated below:

line 33

„cyanobacteriaf” should be corrected to „cyanobacteria”

line 83

the form of indication „°C” is unusual, please check the font(s)

line 86

„Fist” should be corrected to „First”

The manuscript should be accepted after the above minor changes.

Author Response

We are very grateful to a reviewer for the encouraging interest to our research and high opinion of our results. We corrected the typos mentioned.

Reviewer 2 Report

Amina Daminova and colleagues present an article on the structure and chemical properties of melanized thalli in Lombaria pulmonaria by using light microscopy, SEM, TEM and EDS analysis. In general, the article is carefully and comprehensibly written and although mostly descriptive, the conclusions are mainly supported by the results shown and it contains some interesting information for people in this field.

I have the following comments for the authors:

In Figures 3 and 4 the granular structures shown in the melanized sections are described as melanin-like granules. Although the pictures are quite convincing, in my opinion, the authors should be careful with the choice of words used there and in other places of the text, because they have no direct proof that these structures actually represent melanin granules.

Some additional information on the EDS analysis would be welcomed in the Materials and Methods section.

In my opinion, the text would benefit from some enrichment in relevant literature, thus authors should make an effort to search for, incorporate and discuss at least a few additional references in this field. Some examples could be the following (DOI: 10.3389/fmicb.2022.876611, DOI: 10.1186/s12866-022-02505-1)

Author Response

I have the following comments for the authors:

In Figures 3 and 4 the granular structures shown in the melanized sections are described as melanin-like granules. Although the pictures are quite convincing, in my opinion, the authors should be careful with the choice of words used there and in other places of the text, because they have no direct proof that these structures actually represent melanin granules.

Indeed, we were able to visualize such granules in melanized thalli. Using EDX we found that the elemental composition of granules suggests their melanic nature. We certainly agree that more work is needed to confirm this directly, and therefore in our revised m/s we use the words “melanin-like granules” to describe these structures. Moreover, in Fig. 4 we added a scanned image of extracted and purified melanin from lichen Lobaria pulmonaria with EDX spectrum.

Some additional information on the EDS analysis would be welcomed in the Materials and Methods section.

The morphology of the structured surface of the non-melanized and melanized lichen thalli was observed using a high-resolution scanning electron microscope (SEM) Merlin (Carl Zeiss, Germany) at an accelerating voltage of 5 kV, and the elemental composition of melanized samples was determined using an Energy-dispersive X-ray (EDX) detector AZtec X-max (Oxford Instruments, UK) fitted to the microscope at an accelerating voltage of 20 kV. The line intensities for each element of the sample and for the same elements in the calibration standards with known composition were measured [18]. Up to 500,000 impulses were accumulated. The results were normalized up to 100%.

In my opinion, the text would benefit from some enrichment in relevant literature, thus authors should make an effort to search for, incorporate and discuss at least a few additional references in this field. Some examples could be the following (DOI: 10.3389/fmicb.2022.876611, DOI: 10.1186/s12866-022-02505-1)

We added more references and literature information.

Reviewer 3 Report

Dear Authors,

This is a very interesting work exploring the microstructure in the lichen Lobaria pulmonaria in relation to the effect of melanization. You adopted several methods, including reflectance measurements, light microscopy, SEM, and TEM, and the paper reports some very interesting images. They provide also the results on the elemental composition of the cortical layer (EDX).
The article is in general well-written, with an explicative introduction. M&Ms are well described.
I really appreciated the reading of this article and I strongly recommend its publication on JoF.

I have only some minor remarks and suggestions:

- In line 33 there is a typo: ‘(algae and cyanobacteriaf)’ —> ‘(algae and cyanobacteria)’
- line 49 ‘but also their ultrastructure’ —> ‘but also on their ultrastructure’
- line 86 ‘Fist’ —> ‘First’

Discussion.
- Lines 195-196 but also Lines 218 and 2019. In several parts of this section, you comment on the fact that: ‘the high N content of the hyphae measured using EDX suggested n this lichen species that the type of melanin synthesized is eumelanin.’ and ‘The C to N ratio in the melanin granules (Figure 4 B) was c. 11 (Table 1), suggesting that the type of melanin synthesized by this lichen is eumelanin [24]’ Please provide a more detailed explanation for this result. How do you deduce that it is precisely this polymer?

Author Response

- In line 33 there is a typo: ‘(algae and cyanobacteriaf)’ —> ‘(algae and cyanobacteria)’
- line 49 ‘but also their ultrastructure’ —> ‘but also on their ultrastructure’
- line 86 ‘Fist’ —> ‘First’

Typos have been corrected.

Discussion.
- Lines 195-196 but also Lines 218 and 2019. In several parts of this section, you comment on the fact that: ‘the high N content of the hyphae measured using EDX suggested n this lichen species that the type of melanin synthesized is eumelanin.’ and ‘The C to N ratio in the melanin granules (Figure 4 B) was c. 11 (Table 1), suggesting that the type of melanin synthesized by this lichen is eumelanin [24]’ Please provide a more detailed explanation for this result. How do you deduce that it is precisely this polymer?

Several types of melanins, such as eumelanin, allomelanin (or DHN melanin), and pheomelanin, are known to exist. These melanins differ in their elemental composition. In lichens, eumelanin and allomelanin are usually synthesized via different metabolic pathways. Eumelanin has N, while allomelanin almost lacks N, therefore C/N ratio is typically used to distinguish these melanins. Comparison of scanned images of pale and melanized lichen thalli demonstrated the presence of melanin-like granules in melanized thallus only. Furthermore, as a part of our complex study, we extracted and purified melanin from lichen Lobaria pulmonaria and in Fig. 4 we added a scanned image of this melanin with an EDX spectrum.

Reviewer 4 Report

REVIEW OF THE ARTICLE BY AMINA DAMINOVA ET AL. ENTITLED “EFFECT OF MELANIZATION ON THALLUS MICROSTRUCTURE IN THE LICHEN 2 LOBARIA PULMONARIA” (jof-1770564)

The work aims to describe melanization in the cortical layer of the foliose lichen Lobaria pulmonaria (L.) Hoffm. (Lobariaceae). The authors studied melanized and non-melanized thalli mainly by means of microscopy (light, SEM, TEM, X-rays microspectroscopy of SEM images). Reflectance spectra were also analyzed in terms of a “Browning index”. Although the article is in scope of the journal and the topic is interesting, there are very serious concerns related to all parts of the article (Introduction, Methods, Results and Discussion). Main drawback is that the work is similar to previously published papers on Lobaria, which are not cited in the text (Mafole et al. (2019a) Photosynthetica 57(1):96–102; McEvoy et al. (2007) New Phytol 175(2):271–282); Matee et al. (2016) Lichenologist 48(4):311–322; Mafole et al. (2019) Fungal Biol Rev 33(3–4):159–165; Gauslaa , Solhaug KA (2001) Oecologia 126(4):462–471). What is really new? In addition, the paper is not carefully edited according to journal rules (names of divises, English, numbering of subsections, etc.).

Introduction and Discussion. In general, the number of references (29) is too low. Text should be better referenced. Scientific background for melanization in L. pumonaria are not provided, whereas such works exist (Mafole et al. (2019a) Photosynthetica 57(1):96–102; McEvoy et al. (2007) New Phytol 175(2):271–282); Matee et al. (2016) Lichenologist 48(4):311–322; Mafole et al. (2019) Fungal Biol Rev 33(3–4):159–165; Gauslaa , Solhaug KA (2001) Oecologia 126(4):462–471). The same is true about common aspects of melanization in foliose lichenes, exampels of melanins, comparing of brauning index values, patterns of melanization (Rassabina et al. (2020) Biochem Mosc 85:623–628; Gessler et al. (2014) Appl Biochem Microbiol 50(2):105–113; Chekanov et al. (2017). Physiologia plantarum, 160(3), 328-338; Chekanov, Lobakova, E. (2021) Photosynthesis research, 149(3), 289-301). Even main characteristics of the object of the work are not described, i.e. thallus organization, type of phycobiont… There is no conjunction of the work with the current knowledge in lichenology: unfortunatly, the authors do not sue common terms and concepts, do not cite common authors and works, e.g. Büdel Scheidegger (2008) In: Nash TH (ed) Lichen biology. Cambrige University Press, Cambridge, pp 40–68; Honegger R (2008) Morphogenesis. In: Nash TH (ed) Lichen biology. Cambrige University Press, Cambridge, pp 69–93; Honegger R (2012) The symbiotic phenotype of lichen-forming ascomycetes and their endo- and epibionts. In: Hock B (ed) Fungal associations, the mycota IX, 2nd edn. Springer, Berlin, pp 287–339).

Materials and methods. Selected methods and experimental strategy do not correspond to goals. They do not allow us to make most of the conclusions. Although biochemical staining is well-done, nothing was done to determine the chemical nature of “melanin granules”. EDX does not provide absolute values about mass fractions of elements, element analysis by burning is required. It is also very important to select the pattern of the tallus to study melanization (Chekanov, Lobakova, E. (2021) Photosynthesis research, 149(3), 289-301), but it has not been discussed.

Results. Presentation of results is substandard. (1) No statistical treatment for numerical data, no information about number of images for morphometry, (2) EDXR spectra and reflectance spectra are not presented, (3) obtained images are poorly described in the text.

Concluding, the work is weak. However, the data on biochemical staining are nice and could be enhanced and published in a more specialized journal. In order to complete the review, I provide specific suggestions for improving the paper. I believe it will help the Authors for purther publishing.

LIST OF SPECIFIC SUGGESTIONS

  1. l. 3. Add authority to the object of the study: Lobaria pulmonaria (L.) Hoffm.

  2. l. 10. “Lichens are extremophilic organisms” - what do you mean? Although lichens are able to exist under adverse conditions, they are not extremophiles. Many lichens, including Lobaria, live under conditions favorable for most organisms. 

  3. l. 11. “UV light” - UV is not a light. because most of the UV range is invisible. Do You mean “UV radiation”?

  4. l. 12, 202, 256. “s of free-living fungi” - change to “non-lichenized fungi”, because melanization is also a common feature of parasitic ones.

  5. l. 10-13. The text should be modulated here. Not all lichens have a differentiated thallus with upper cortex. What is about lichens without dorsoventral differentiation, about microlichenes fruticose lichens?.. In your work the attention is paid only to a case study, i.e. foliose lichen with the heteromeric dorsoventral thallus.

  6. l. 15. “d cells sensitive to phenol” - what do you mean? fungal cells? The tallus of L. pulmonaria does not consist from cells it is represented by hyphae. 

  7. l. 10-23. Remove unused abbreviations in the Abstract (SEM, TEM). At the same time DOPA and EDX have to be written in full.

  8. l. 28. Lichens are rather supraorganismal symbiotic systems, than “photosynthetic organisms”.

  9. l. 28-29. Add reference.

  10. l. 35. UV is not a light.

  11. Add references to introduction to common works on melanization in lichenes, particularly, in L. pulmonaria.
  12. l. 33. typing error

  13. l. 33. “algae or cyanobacteriaf” - change to “green algae and/or cyanobacteria)”.

  14. l. 38. I am not sure about “all kingdoms”. Particularly, Chromalveolates and Rhodophytes do not exhibit melanin. Moreover, I have not found such information in your ref. [5].

  15. l. 52. Add authority at the first mention.

  16. l. 58-62. Please, add reference(s).

  17. In the Materials and Methods subsections should be italicized and numbered (please, see journal template). “Materials” - should be “Lichen material”. Remove the word “Methods”.

  18. l. 72. Dried at which humidity?

  19. l. 68-73. Please, specify the conditions in more detail: geographical coordinates, date, time of the year, weather.

  20. l. 76. What are “lichen discs”? Fragments of lichen thalli?

  21. l. 76-81. Description of the procedure is unsatisfactory. “Browning indexes of the lichen thalli were assessed using an integrating sphere” - what do you mean?? Do you mean, that reflectanse spectra in the range of  400-1050 nm were registered on a Dual Channel SD2000 Miniature Fiber Optic Spectrometer equipped with an integrative sphere ISP-50-REFL. Provide the diameter of the sphere. where was it located?

  22. l. Browning indexes - what is it? Do you mean the Browning Reflectance Index (BRI)? If yes, please, provide the expression for calculations, and original reference to BRI (Chivkunova et al. (2001) J Russ Phytopathol Soc 2:73–77).

  23. l. 67-121. According to rules of the journal, countries of origin and cities should be indicated for all manufacturers.

  24. l. 83. At which humidity?

  25. l. 91. 50:50 by volume?

  26. l. 95. 2.5% v/v?

  27. l. 96. Na-phosphate?

  28. l. 98. f 1:3, 1:1, 3:1 - by volume?

  29. l. 97. If you dehydrated your samples by 96% ethanol only, you did not completely remove the water. You must use 100% ethanol at the final stage.

  30. l. 104. WAS determined.

  31. l. 104. EDX has to be described.

  32. l. 95-107. How did you dry your samples?

  33. l. 105-107. I am not sure, that it is correct to measure cell wall thickness on SEM images, because you do not take into account the layer of the metal. Moreover, You samples were not dried at a CO2 critical point, thus, these measurements could be an artifact of the experiment.

  34. l. 110. a-phosphate?

  35. l. 112. 1% osmium tetroxide - where? in a buffer? 1% wt/v?

  36. l. 118. Ultracut III - manufacturer and place of origin?

  37. l. 199. Provide the reference for Reynolds' lead citrate (Reynolds (1963) The Journal of Cell Biology, 17(1), 208.).

  38. Results must be divided into subsections.

  39. l. 125. “specific layer of pigmented cells” - I do not see “cells” on Figure 1. Moreover there is no fungal cells in lichen thalli of L. pulmonaria

  40. l. 126. “thickness of pigmented layer was on the 126 average 20 µm” - please add statistics. present data oas averages and errors. From which number of crossections?

  41. It is very important to indicate which part of the thallus you use in the study in methods and results (for details, please, see https://doi.org/10.1007/s11120-021-00860-0).

  42. On Figure 1, please, indicate the main parts of the thallus: upper cortex, layer of phycobiont, medulla, melanized zone. 

  43. When describe thallus cross sections, please describe it using common terms of lichen anatomy (Honegger R (2008) Morphogenesis. In: Nash TH (ed) Lichen biology. Cambridge University Press, Cambridge, pp 69–93; Büdel B, Scheidegger C (2008) Thallus morphology and anatomy. In: Nash TH (ed) Lichen biology. Cambrige University Press, Cambridge, pp 40–68).

  44. l. 137-138. Description is insufficient. Please, at least indicate obtained values in the text. These numerical data should be treated: what do values and errors mean (Figure 3)? It should be indicated. How many replicates? Were the differences statistically significant? How did you determine it? Indicate difference by symbols above bars.

  45. l. 142-150. Please, provide a general description of thallus ultrastructure in this zone using common terms of lichenology. Use terms paraplectenchyme or prosoplectenchyme.

  46. Figure 3 (caption) - what do arrows mean?

  47. l. 146. “melanin-like granules” - in my opinion, it is bacterial cells.

  48. l. 149. Provide these values in the form of a table. Describe statistics: from how many images? Provide average density of hyphae per, e.g. 100 1 µm and standard deviations or standard errors.

  49. Add representative reflectance spectra for BRI calculation.

  50. Add representative EDX spectra for analyzing of element composition as it is commonly accepted in the field.

  51. l. 169. How did you determine mass fractions of elements by EDX (Table 1)? As far as I know, it is impossible without a calibration. You should use an element analyzator. From EDX spectra you can calculate only areas under the peaks corresponding to elements, but it is not proportional to mass fractions of elements. 

  52. l. 172-178. Please, describe cross-section using common terms of lichen ultrastructure.

  53. l. 172-178. I do not think that this is solid evidence, that these osmiophilic granules are of melanin nature. You should at least compare TEM images of melanized and non-melanized cortex. I do not see melanin vesicles on Figure 5C,D (as in the caption). Most likely, it is cell walls.

  54. l. 187. What is “ morphology of lichen melanins”? Melanin is a molecule. Do you mean its chemical structure?

  55. l. 193-195. High N content also could be attributed to chitin cell walls of fungi.

  56. l. 196. These changes are rather anatomical than morphological.

  57. l. 199-200. This description does not correspond to common terms in lichen anatomy. Please, modulate it accordingly (Honegger R (2008) Morphogenesis. In: Nash TH (ed) Lichen biology. Cambridge University Press, Cambridge, pp 69–93; Büdel B, Scheidegger C (2008) Thallus morphology and anatomy. In: Nash TH (ed) Lichen biology. Cambrige University Press, Cambridge, pp 40–68). Use the terms aerial hyphae, medulla, prosoplectenchyme…

  58. l. 207-217. It is not “melanin ultrastructure”, but the form of melanin depositing.

  59. l. 220-221. “CHNS-O Elemental Analyzer 220 (Eurovector SpA, Italy)” - methods.

  60. l. 259. Pathogenic fungi are not free living. They are free-living and parasitic.

Round 2

Reviewer 4 Report

I have clearly read the responces and revised text. Unfortunatly, the Authors provided just formal answers to the commentsl although editorial decision was "Major revision", no signifficant changes done in the revised version. Even subsections were not numbered and authors of the species were not added to the title. Some poits mentioned in the responses as corrected were not actually corrected in the text. Again I provide main comments to the work.

1) According to authors’ guidelines, “...The introduction should briefly place the study in a broad context and highlight why it is important… The current state of the research field should be reviewed carefully and key publications cited… Keep the introduction comprehensible to scientists working outside the topic of the paper”. In current state the Introduction does not fit this definition. A matter of the study is the foliose lichen Lobaria pulmonaria (L.) Hoffm. It is not clear for readers not specialized in the biology of this species: at least type of differentiation of its thallus (dorsoventral, foliose, heteromeric), main characteristic of each lichen is its main components (is it two- or three component, are there cyanobacteria, green algae)... Latest is especially important, because it exhibits cephalodia, and a reader should understand what he/she sees on cross sections. Were there cephalodia-containing fragments or not. Unfortunately, authors do not think that it is important. Then, the authors write that they “ have discussed the physiological and biochemical characteristics of melanisation in our earlier comprehensive review and numerous original articles”. This response is unacceptable, because it was not done in CURRENT WORK. The authors describe melanization in L. pulmonaria, possible chemical structure of its melanin, BRI values. Such data, which are very close to current work, is not elucidated. Thus, in the current state, it makes a feeling that this data is not discussed in detail to artificially increase the novelty of the study.

2) According to authors’ guidelines and general principles of scientific writing, “Authors should discuss the results and how they can be interpreted in perspective of previous studies and of the working hypotheses…”. It has not been done, despite repeating some measurements from previous work, particularly, they did not compare their BRI values for Lobaria with that from previous works on the same species and other close-related foliose lichens (i.e. from the references Mafole et al. (2019) Photosynthetica 57(1):96–102; McEvoy et al. (2007) New Phytol 175(2):271–282; Matee et al. (2016) Lichenologist 48(4):311–322; Chekanov, Lobakova, E. (2021) Photosynthesis research, 149(3), 289-301). The authors describe patterns of melanization in L. pulmonaria. Similar Studies were done on the same species in the work McEvoy et al. (2007) New Phytol 175(2):271–282; Gauslaa , Solhaug KA (2001) Oecologia 126(4):462–471, but they do not mention this work and do not say whether these results are comparable. Why? To artificially increase novelty of the study? One can see an average number of references in the articles in the field, and especially in the Journal of Fungi. It is higher than in the presented paper.

3) In descriptive biological works it is necessary to provide a correct and detailed description of the object in your studies. It is not the case of the current text. General morphology of selected thalli (shape, thaline differentiation, size) and description of anatomy on cross sections  are not presented. It is important, at least for understanding, whether the Authors describe normal or pathological thalli, where they completely developed. For understanding of melanization patterns, it is very important to select the zone of the thallus and its side (I provided works on this issue - Chekanov et al. (2017). Physiologia plantarum, 160(3), 328-338; Chekanov, Lobakova, E. (2021) Photosynthesis research, 149(3), 289-301). Was the developed thally studied? To understand it, they should indicate on cross sections features of developed thallus zone by common lichenological terms (upper, lower cortex, rhizomes and rhizoids, cephalodia, medulla, layer of phycobiont…) in accordance with common works (. Büdel Scheidegger (2008) In: Nash TH (ed) Lichen biology. Cambridge University Press, Cambridge, pp 40–68; Honegger R (2008) Morphogenesis. In: Nash TH (ed) Lichen biology. Cambridge University Press, Cambridge, pp 69–93; Honegger R (2012) The symbiotic phenotype of lichen-forming ascomycetes and their endo- and epibionts. In: Hock B (ed) Fungal associations, the mycota IX, 2nd edn. Springer, Berlin, pp 287–339). It is also to use and understand the terms basal and apical zone, because it dictates degree of melanization. The Authors did not follow this suggestion. Even in the BRI measurements, they did not pay attention to what size of thallus they studied. It was done in all previous works (e.g. Mafole et al. (2019); McEvoy et al. (2007) New Phytol 175(2):271–282; Matee et al. 2016; Chekanov and Lobakova, 2021).

4) Discussion of the data of electron microscopy suffers from overinterpretation. Let us first consider SEM and TEM data. First of all, we cannot conclude about the nature of the presented granules. Indeed, these structures have a similar features with that reported in Camacho et al. (2019) cited in the text, but in that work the  Camacho et al. use other spectroscopic methods to verify their possible chemical nature to provide solid evidence. By contrast, there are many works, where similar structures correspond to metal nanoparticles or coccoid bacterial cells (see e.g. Chekanov et al. (2017). Physiologia plantarum, 160(3), 328-338; Honegger, R. (2009). In Plant relationships (pp. 307-333). Springer, Berlin, Heidelberg). The authors could at least compare the diameter of granules with data by Camacho et al. and determine thickness of a putative “melanin layer” of the cell wall. The same is true about the data of element analysis. They verify melanin's nature by the fact that the samples contained N. Many fungal structures contained N. To say about melanins, it is important to cmpare N fraction and/or C/N ratio with that for other melanins. I do not take into account Beilinson, et al. (2022), because adding references to unreviewed papers is unacceptable. At the same time there are previously published works, where the chemical nature of Lobaria melanins was shown more accurately (this is a question about novelty). In my point of view, it is also important to add the fact, that L. pulmonaria contains N-fixing cyanobacteria, which is a feature of lichens with eumelanin (Mafole et al. (2019) Photosynthetica 57(1):96–102; McEvoy et al. (2007) New Phytol 175(2):271–282; Matee et al. (2016) Lichenologist 48(4):311–322; Chekanov, Lobakova, E. (2021) Photosynthesis research, 149(3), 289-301). It reflects the absence of novelty. In general, spectroscopic data, i.e. are required to show chemical structure (Rassabina et al. (2020) Biochem Mosc 85:623–628). The authors even did not follow my suggestion to show reflectance spectra to show typical features of melanin-containing samples. Considering TEM data, in vitro studies of vesicle migration are required to support the hypothesis about intracellular changes during thallus melanization or at least references. New data on purification of putative melanins were added. Why were not SEM images of “purified melanins” similar to “melanin granules”? What was done to verify the chemical nature of “purified melanins”, e.g. absorbance spectra?

5) Presentation of the EDX spectrum is substandard. Quality of the figure is low. I suggest discussing with experts in this field how to analyze and to present this sort of data.

I shall not dwell, on more specific but important issue, e.g. that in the “expression for BRI calculation” it is not explained, what are R550, R700, and R750,  and that it is not an expression for BRI calculation, because it must contain a symbol “=” and BRI, names of axes must be on all graphs, etc.

Concluding, I retain my original suggestions of the work. I do not suggest SEM and TEM data for publication. At the same time, the Authors did good work on biochemical staining of L. pulmonaria thalli. It could be published in a more specialized journal after improvement, however I see that they do not want to improve the text.
